Review

 

Subject Area:
biochemistry/biophysics/cellular biology

Keywords:
ATP synthesis, mitochondria, oxydative phosphorylation, chemiosmotic theory, $F_oF_1$-ATP synthase

Author for correspondence:
Alessandro Maria Morelli
e-mail: morellia@unige.it

# An update of the chemiosmotic theory as suggested by possible proton currents inside the coupling membrane

Alessandro Maria Morelli[1], Silvia Ravera[2], Daniela Calzia[1] and Isabella Panfoli[2]

[1]Pharmacy Department, Biochemistry Lab, University of Genova, Viale Benedetto XV 3, 16132 Genova, Italy
[2]Experimental Medicine Department, University of Genova, Via De Toni 14, 16132 Genova, Italy

AMM, 0000-0002-4625-5223; SR, 0000-0002-0803-1042; DC, 0000-0002-9885-720X;
IP, 0000-0002-6261-1128

Understanding how biological systems convert and store energy is a primary purpose of basic research. However, despite Mitchell's chemiosmotic theory, we are far from the complete description of basic processes such as oxidative phosphorylation (OXPHOS) and photosynthesis. After more than half a century, the chemiosmotic theory may need updating, thanks to the latest structural data on respiratory chain complexes. In particular, up-to date technologies, such as those using fluorescence indicators following proton displacements, have shown that proton translocation is lateral rather than transversal with respect to the coupling membrane. Furthermore, the definition of the physical species involved in the transfer (proton, hydroxonium ion or proton currents) is still an unresolved issue, even though the latest acquisitions support the idea that protonic currents, difficult to measure, are involved. Moreover, $F_oF_1$-ATP synthase ubiquitous motor enzyme has the peculiarity (unlike most enzymes) of affecting the thermodynamic equilibrium of ATP synthesis. It seems that the concept of diffusion of the proton charge expressed more than two centuries ago by Theodor von Grotthuss is to be taken into consideration to resolve these issues. All these uncertainties remind us that also in biology it is necessary to consider the Heisenberg indeterminacy principle, which sets limits to analytical questions.

## 1. Introduction

The 'chemiosmotic theory' formulated by Mitchell [1], a researcher with an Anglo-Saxon training in chemistry, dates back more than 50 years. The theory has universally been accepted, although it immediately raised several controversies, which lasted until today. An upgrading of the chemiosmotic theory appears necessary in light of the enormous progress of bioanalytic techniques defining the fine structure of the macromolecular complexes involved in oxidative phosphorylation (OXPHOS) [2–7], notably studies on complex I (NADH: ubiquinone oxidoreductase) and complex IV (cytochrome *c* oxidase) [2,3,8]. These data allow further insight into the proton pathway, a key issue of the theory. The actual proton path across the membrane, the putative proton concentrations on either sides of the membrane and the consequent membrane potential have been the subjects of countless studies. In all evidence, it appears that a free proton osmosis would be impossible, it being a quantum particle that binds to water forming hydronium ions ($H_3O^+$). In fact, any free proton in the membrane would quickly be drained by the aqueous phase, releasing the energy associated with the solvation process, to the detriment of the membrane. Also, free protons display a huge destructive force on any biological membrane they pass through. Accordingly, some membrane transporters (such as the potential-dependent proton pump Hv1) are designed specifically to prevent the proton destructive force [9]. Also, a number of reports

royalsocietypublishing.org/journal/rsob    Open Biol. **9**: 180221

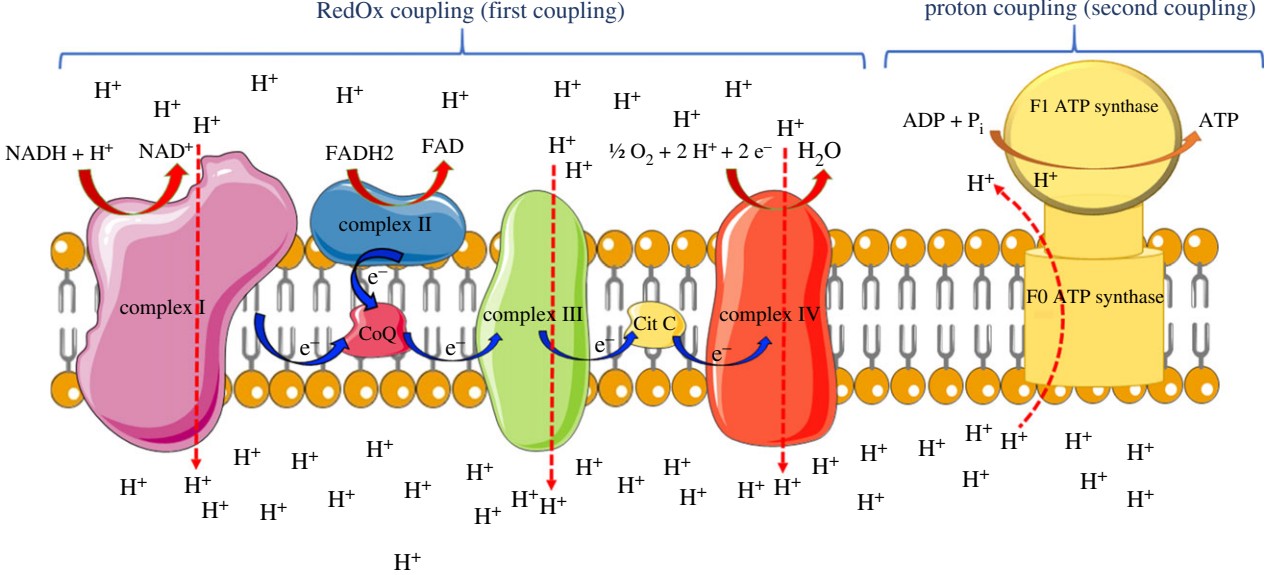

matrix side or N side

RedOx coupling (first coupling)

proton coupling (second coupling)

periplasmatic side or p side

**Figure 1.** Schematic of the 1961 Mitchell chemiosmotic theory. A delocalized coupling is depicted among protons extruded by the electron transport chain (ETC) and ATP synthesis. The overall process is arbitrarily divided in the two phases: the 'RedOx coupling', in which the proton movement is operated by the ETC, and the 'proton coupling', in which proton movement is coupled with ATP synthesis, by $F_oF_1$-ATP synthase.

argue that protons accumulating onto the respiring membrane never reside in the aqueous phase [10–13].

Therefore, in this work, we hypothesize that the coupling modality between the electron transport chain and the protonic movement could happen inside the membrane to prevent a proton release, opening new scenarios to explain the basic mechanisms of aerobic metabolism. In other words, in this review, we debate the possibility to update the chemiosmotic theory and unravel the role of local processes in the coupling. This may help in developing new strategies for innovative research centred on cellular bioenergetics.

## 2. The chemiosmotic theory and $F_oF_1$-ATP synthase

The basic formulation of Mitchell's theory is schematically depicted in figure 1, where ATP synthase was also indicated, differently from the original release of 1961, where necessarily it was not depicted. At the time ATP synthesis was attributed to the membrane as a whole, in the form of a generic subtraction of $H^+$ and $OH^-$ to ADP and orthophosphate to form ATP.

The experimental data in support of the theory came successively and are reported in literature as a huge amount of contributions. Reviews have been published and we refer to them for a complete documentation [14,15], only the most significant issues being mentioned here. Particularly influential were the data produced in 1966 by A. T. Jagendorf & E. Uribe in the famous 'acid bath experiment' [16]. They obtained an ATP synthesis inducing a transmembrane leap of pH in chloroplasts *in vitro*. In the same year, Y. Kagawa & E. Racker ascertained that the synthesis of ATP occurred on the so-called 'spheres' referred to as $F_1$ subunits of the ATP synthase [17]. Since then the basic contribution of $F_oF_1$-ATP synthase (ATP synthase) to the OXPHOS became clear.

Developments in the molecular knowledge regarding ATP synthase have been comprehensively addressed in many reviews [18–20]. In summary, we can say that the theory is based on three basic postulates:

(1) an electron transport chain, providing the energy for $H^+$ transfer from one side to other side of the membrane;
(2) ATP synthase, synthesizing/hydrolysing ATP through $H^+$ translocation; and
(3) impermeability of the inner mitochondrial membrane to ionic species thereby including protons.

The basic requirement for the OXPHOS is a coupling between redox processes, proton translocation and ATP synthesis. Such global coupling can arbitrarily be divided into two distinct phases: a coupling between the oxidation-reductive process and the protonic translocation, referred as 'RedOx coupling' or 'first coupling' (see figure 1, left; see also recent review in [21]) and the coupling between protons accumulated on the p-side of the membrane moving to the n-side through the ATP synthase, which determines the synthesis of ATP, here referred as 'proton coupling' or 'second coupling' (see right side of figure 1). Considerable attention was devoted in the 1980s and 1990s to clarifying the structural–functional details of the respiratory complexes (I, II, III and IV) and ATP synthase (complex V). Important was the study of respiratory complexes organized in supercomplexes [22–24] with the demonstration that the loss of their aggregation leads to an increase in the production of reactive oxygen species [24,25]. The possible participation of complex V to supercomplexes has never been demonstrated [23]. The study of supercomplexes has also benefited from extraordinary surveys carried out on X-rays [26].

By contrast, the 'proton coupling' or 'second coupling' (see figure 1, right) appears to be the most critical passage of the whole OXPHOS process. Literature reports several experiments performed with reconstructed systems [27,28] (i.e.

ATP synthase incorporated into phospholipid vesicles) carried out about fifteen years after Mitchell's hypothesis. Vesicles obtained from the membranes from the purple *Halobacterium salinarum* synthesized ATP as a result of illumination, and it was thus demonstrated that proton movements through the membrane support the synthesis of ATP. In 1977, N. Sone *et al.* observed that ATP synthase purified by *Thermophilic bacterium*, inserted in artificial membranes, was able to synthesize ATP thanks to a transient shift in membrane potential ($\Delta\Psi$) induced by valinomycin, allowing rapid passage of $K^+$ ions across a membrane on the sides of which different salt concentrations were set [28]. These experiments demonstrated that proton translocation is the crucial step for the 'proton coupling' between protonic movement and ATP synthesis. On this general topic, pivotal is the minireview by Junge [29, p. 197], which on one hand highlights the versatility of the ATP synthase, a nano-machine 'unique in converting electrochemical, mechanical and chemical forms of energy' and on the other hand points out that there is still much to be understood about the chemical–physical basis of such process.

# 3. Controversies about the chemiosmotic theory

A long struggle was necessary for the chemiosmotic theory formulated by Mitchell [1] to be widely accepted. The controversy, central to the history of bioenergetics for more than half a century, appears tackled by more than 200 articles and to have lasted until the most recent years. G. F. Azzone, many years ago (1972), published the manuscript 'Oxidative phosphorylation, a history of unsuccessful attempts: is it only an experimental problem?' [30], which already highlighted the non-convincing parts of the theory, wishing for answers from the fine analysis of the macromolecular structures involved in chemiosmosis.

The harshest criticisms came from J. Prebble, who emphasized the lack of experimental data in support of the theory [31]. W. Junge effectively described, in his recent review 'Half a century of molecular bioenergetics', the chronicle of the dispute, which even took harsh tones [32]. S. Brown & D. C. Simcook [33] considered the motivations that have convinced the scientific community to accept the chemiosmotic theory. The authors note that: 'science shows tremendous resistance to change and it takes extraordinary perseverance to persuade the community' [33, p. 178].

A controversial issue was the correlation between the membrane potential ($\Delta\Psi$) and the protonmotive force, often considered equivalent entities [34]. It was postulated that a $\Delta\Psi$ with positive charge on the external p-side of the internal mitochondrial membrane and negative on the n-side in contact with mitochondrial matrix (figure 1) would let protons enter the $F_o$ rotor that synthesizes ATP in the matrix thanks to its mechanical connection with the $F_1$ moiety. Protons would gather across the coupling membrane like chemical ions, creating a driving force for $F_oF_1$-ATP synthase to synthesize ATP, realizing the 'proton coupling'. However, the yield in ATP poorly correlates with bulk-to-bulk membrane potential [35,36], questioning the basics of chemiosmotic theory [37].

The award of the 1978 Nobel Prize for chemistry to Peter Mitchell cooled the dispute, but not definitively. In fact, in 1979, there was a heated confrontation published in *Trends*

*in Biochemical Sciences* between H. Tedeschi, who disproved the idea that the metabolic activity of mitochondria could contribute to membrane potential, and H. Rottenberg, which instead defended Mitchell's theory [38]. The original theory provides a protonated 'delocalized coupling' (as depicted in figure 1), as opposed to a 'localized coupling' supported by Williams [39]. Lee [40] reported a rigorous chemical/physical experiment in favour of localized coupling, demonstrating furthermore that the thylakoid membrane can be a 'proton capacitor'. The putative existence of a proton capacitor is a matter of great importance, and later, Saeed & Lee [10] showed that protons can actually accumulate on the membrane surface even though they never reside in the aqueous phase. Moreover, concerning the experimental verification of the 'proton coupling', a recent elegant investigation in HeLa cells, bioengineered with green fluorescent protein as pH indicator inserted in respiratory complex III and in $F_o$ moiety of ATP synthase, points to a localized coupling [41].

A report entitled 'Proton migration along the membrane surface and retarded surface to bulk transfer' by Heberle *et al.* [11] interestingly reconciles the two visions, providing proof that proton transfer from a proton generator (bacteriorhodopsin) to an acceptor (water-soluble pH indicators) is faster if occurring on the membrane rather than when protons are released in the aqueous bulk. Ferguson [12] emphasized Heberle *et al.*'s experiments, concluding that the delocalized coupling and lateral proton transfer (localized coupling) between the proton generator and user occurs very rapidly on the membrane, as compared with the slower and transversal passage through the aqueous bulk. In this context, the recent paper from C. von Ballmoos's group observed that $\Delta\Psi$ and $\Delta$pH are equivalent for the coupling with ATP synthase [34]. A primary role for membrane buffering on proton mobility in general can be hypothesized [42]. The experimental data [43,44] showing a close thermodynamic correlation between valinomycin-induced $\Delta\Psi$ and ATP synthesis in reconstituted systems are very important, but it seems plausible that they induce a transmembrane protonic flow that probably differs from the path in a native environment.

Moreover, the eminent English chemist R. J. Williams clearly rejected the hypothesis of the accumulation of protons from p-side: 'the p-phase corresponds to the infinitely extended external space. If protons are extruded into this "Pacific Ocean", they would be diluted and the entropic component of the pmf would be lost' [13, p. 18]. R. Williams observed that 'protons in the membrane rather than an osmotic trans-membrane gradient of protons were required to drive ATP formation', based on a series of considerations that excluded the presence of free protons from p-side [39, p. 123]. An elegant demonstration of Williams's localized coupling hypothesis came in 1976 [45] by an experiment in which purified ATP synthase was added to the octane–water interface. It was observed that protons accumulate in octane, a Brønsted acid, leading to ATP synthesis by ATP synthase. These data have also been recently confirmed [46]. Eighteen years later, it was reported that 'our results suggest that protons can efficiently diffuse along the membrane surface between a source and a sink (for example H+-ATP synthase) without dissipation losses into the aqueous bulk' [11, p. 379]. From all the cited data it can be concluded that protons (or more likely protonic currents) are confined into the membrane, while proton exit from the

royalsocietypublishing.org/journal/rsob    Open Biol. 9: 180221

royalsocietypublishing.org/journal/rsob   Open Biol. **9**: 180221

membrane is to be considered only as a fallback way of escape, mostly via *in vitro* reconstituted conditions.

A direct measurement of membrane potential of the mitochondrial inner membrane with microelectrodes was only be accomplished by H. Tedeschi, who showed the existence of a positive inside and negative outside mitochondrial $\Delta \Psi$ [47,48]. Such potential (contrary to the canonical one) interestingly coincides with that calculated on the basis of the ionic species present on the membrane sides [49]. Clearly, knowledge of the entity and especially the sign of this potential is fundamental for understanding the basic functioning of chemiosmosis, as emerges from the already mentioned historical dispute between Tedeschi and Rottenberg [38].

To measure $\Delta \Psi$, laboratory tests currently use lipophilic fluorescent compounds whose response is considered to be related to $\Delta \Psi$. However, tests conducted with rhodamine have cast doubts on such correlation [50] since these indicators inhibited the mitochondrial respiration, so they disturb the system. As such compounds dissolve into the membranes, they may reflect the membrane behaviour; in fact they inhibit a membrane intrinsic process (i.e. the OXPHOS), but do not interfere with $\Delta \Psi$. Surely, *in vitro*, proton passage across membranes can be forced with rapid movements of potassium ions by addition of valinomycin [28,51]. A laboratory procedure using valinomycin and also nigericin has been widely used to create a transient $\Delta \Psi$ operating a delocalized coupling linking $\Delta \Psi$ and ATP synthesis, but this does not exclude that in the native membranes a localized coupling would operate independently of $\Delta \Psi$.

## 4. A crucial issue: the membrane permeability to protons

In a previous study (1986), M. Zoratti *et al*. highlighted the uncertainties of the proton cycle [52]. In the same year Grzesiek & Dencher [53] showed that the phospholipid membranes are intrinsically permeable to protons. Data show that phospholipid membranes, normally impermeant to ionic solutes (transversal or permeability coefficients varying between $10^{-12}$ to $10^{-14}$ cm s$^{-1}$), exhibit a significant proton permeability, varying from $10^{-3}$ to $10^{-9}$ cm s$^{-1}$. Such variability may be justified by a buffering capacity of the membranes for protons: proton diffusion value could depend on the higher or lower degree of pre-existing protonation. Recently, the proton leak through lipid bilayers was modelled as a concerted mechanism [53–55]. Tepper & Voth [54] provided a theoretical interpretation of proton permeability, based on the formation of transient membrane spanning aqueous solvent structure. High proton permeability has also been confirmed in liposomes, independently form their phospholipid composition [56]. It is clear that a high degree of permeability to protons is *per se* in contrast to the third of the aforementioned basic postulates of the chemiosmotic theory. With regard to the relationship between the membrane and aqueous phases, many observations confirm the existence of a layer of water molecules on the two sides of the membrane, which to some extent isolates it from the aqueous phases present on its two sides [57–59]. In particular, E. Deplazes *et al*. observed that at the membrane level, $H_3O^+$ forms strong and long-lived hydrogen bonds with the phosphate and carbonyl oxygens in phospholipids [60].

## 5. Proton solvation

A central issue is the actual chemical species of the proton: free, or in the form of $H_3O^+$? This depends on the phase in which the proton is located. Protons possess peculiar chemical properties, being essentially an atomic nucleus. Free protons do not exist in the aqueous phase, being solvated to $H_3O^+$, from which the extraction of a proton would be virtually impossible. In fact, in the transition from $H_3O^+$ to free proton a strong energy barrier higher than 500 meV must be overcome. The desolvation barrier [61] corresponds to the enormous amount of 262 400 Cal mol$^{-1}$ [62]. An immense literature exists on the subject [40,61,63,64]. An interesting report [65], not sufficiently taken into account, calculated the number of free protons (actually in the form of $H_3O^+$) in the volume of a mitochondrion, whose the order of magnitude is femtolitres. Starting from basic physical chemical data (Avogadro number, ionic water product, mathematical pH expression and mitochondrial volume), this study demonstrated that free protons in a mitochondrial periplasmic space are too few (fewer than 10) to support any process dependent on proton translocation in the aqueous bulk across the membrane and absolutely inadequate to support the thousands of ATP synthase molecules present in a mitochondrion. Moreover, the pH value inside the mitochondrion was shown to differ by 0.5 units from what was previously believed [66]. Indeed, the huge energy associated with proton solvation would have a negative consequence: a free membrane proton would quickly be 'sucked' by the near aqueous phase, releasing the huge energy associated with the solvation process, to the detriment of the membrane.

## 6. Grotthuss mechanism and proton translocation through the membranes

A putative mechanism for proton diffusion was hypothesized more than two centuries ago by von Grotthuss [67] and is synthetically explained by S. Serowy and colleagues: 'Proton diffusion according to the Grotthuss mechanism occurs much faster than molecular diffusion because it is uncoupled from the self-diffusion of its mass' [68, p. 1031]. Protons would not diffuse as a mass, rather as a charge, the latter moving between water molecules or protonable groups of suitable macromolecules of the membranes. The Grotthuss mechanism allows better understanding of the possible ways in which protons or species derived from them move in biological systems. T. E. DeCoursey published a review [68] which exhaustively analyses proton transfer pathways in water and biological membranes [9], including Hv1 channels that specifically transfer protons in aqueous phase, therefore actual acidity from one side of the membrane to the other. The mechanism of the voltage gated proton channel Hv1 is not as yet resolved, as emblematically stated by the title of the paper: 'The voltage-gated proton channel: a riddle, wrapped in a mystery, inside an enigma' [69]. Two mechanisms have been proposed for Hv1, as schematically depicted in figure 2. On the left side of figure 2 is the so-called 'frozen water' mechanism, in which the channel traps one or more molecules of water allowing protons to pass through with Grotthuss-style proton hopping, as in the typical case of Gramicidin [70]. On the right side of figure 2 is a passage of protons with protonation/

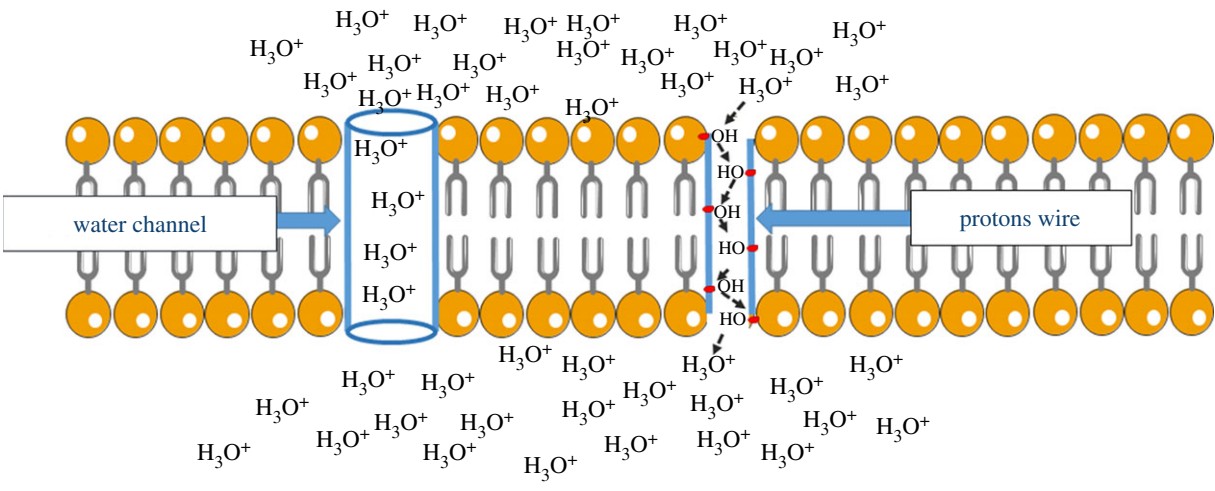

**Figure 2.** Mechanism for H$^+$ transfer through the membrane by H$_V$1. On the left is the water channel model: the water molecules allow protons to pass through with Grotthuss-style H$^+$ hopping. On the right is the proton wire model: a charge migration occurs (through with Grotthuss-style H$^+$ hopping) on polar groups of side chains of amino acids of H$_V$1.

deprotonation of amino acid side-chains, which would realize the so-called 'proton wire' already proposed in the paper by Nagle & Morowitz [71]. The topic was consolidated successively by Nagle & Tristram-Nagle [72]. T. E. DeCoursey in a debate recently published in the *Journal of Physiology* supports the mechanism shown in the right part of figure 2 (i.e. proton wires through the membrane [73]), while Bennett & Ramsey [74] support a mechanism of passage through water molecules as schematized on figure 2, left. Interestingly, both mechanisms are based on the Grotthuss proton movement between (i) water molecules in the case of the water channel (on the left) and (ii) amino acid side chains in the case of protons wires (on the right).

The proton movement in membranes (both biological and artificial) has been analysed by many rigorous chemical/physical studies. The article '"Proton holes" in long-range proton transfer reactions in solution and enzymes: a theoretical analysis' shows that other compounds in addition to water are involved in the 'proton hopping' [75], and is interesting that a quantum-mechanical approach is applied [76]. The article 'Grotthuss mechanisms: from proton transport in proton wires to bioprotonic devices' presents devices such as proton diodes, transistors, memories and transducers, semiconductor electronic devices that use the Grotthuss mechanism [77].

Notably, Hv1 only allows the passage of protons in the form of hydronium ions, balancing their concentration between two aqueous compartments separated by a membrane. This property does depend on the membrane potential $\Delta\Psi$, similarly to other ion membrane devices abundant in biological membranes (for example the Na$^+$ and K$^+$ voltage gated channels), acting on the conformation of Hv1. It appears therefore that to carry protons through the membrane there are *ad hoc* structures that deeply differ from the respiratory complexes, both for their finality and the molecular mechanism. From this and other evidence it can be concluded that the native proton movements in the respiring membranes, when there is no need for acidification of the milieu on one side of a membrane, must take place entirely inside the membrane [78]. As far as the respiratory complexes are concerned, on the other hand, it is theorized that the proton and membrane potential movements are mutually dependent, in that the

pumping of protons would *generate* the membrane potential, which is impossible for thermodynamic considerations that we will elaborate later on.

Moreover, this comparison between proton movement in support of the OXPHOS and the actual protonic movement in nature sheds light on the fact that when protons are really transferred through a membrane (i) they are never in the form of free protons and (ii) they are subject to the Grotthuss-style proton hopping. Hence the need for chemiosmotic theory to be updated in the light of all this emerges.

# 7. Inhibition of ATP hydrolysis: an open topic

ATP synthase inherent catalytic properties differentiate it from the vast majority of other enzymes/catalysts. In fact, it transfers energy to the reaction it catalyses, therefore it influences the equilibrium of the reaction while the other enzymes are irrelevant with respect to it. This peculiar property also requires that the activity of ATP synthase be subjected to proper control, as once the coupling with the oxide/reduction systems is lost, the nanomotor itself rapidly hydrolyses ATP, resulting fatal for cell survival. The significant free energy release ($\Delta G^\circ = -7300$ cal mol$^{-1}$) involved in ATP hydrolysis thus establishes a clear directionality towards the process of hydrolysis itself. It should be emphasized that for most enzymes, the direction of the reaction depends on the concentration of the reactants/substrates, which obviously does not occur for ATP synthase.

To inhibit the reversal of ATP synthase in uncoupled conditions, the action of the inhibitor protein IF1 (UniProtKB: P01096) is essential, particularly in the brain [79], which is relatively devoid of energy reserves (glycogen, triglycerides), where anoxia would lead to the tumultuous ATP hydrolysis. IF1 knock down in HeLa cells [80] equipped with a FRET sensor for measuring the intracellular ATP concentration, surprisingly, did not affect cell viability or mitochondrial morphology, even though the cellular ATP concentration decreased by about a third. Moreover, recently it was reported that deleting the IF1-like $\zeta$ subunit from *Paracoccus denitrificans* has little influence on ATP hydrolysis

royalsocietypublishing.org/journal/rsob    Open Biol. **9**: 180221

by ATP synthase [81], confirming that the inhibition of the hydrolysis of ATP by uncoupled ATP synthase does not depend only on IF1 and that the subject requires further investigations. Also, in bacteria, when the concentration of the ATP reaches the physiological value, the $\varepsilon$ subunit inhibits ATP synthase by binding to the c ring, and it has been proposed as a target for the design of anti-tuberculosis drugs [82].

# 8. Local processes for the coupling inside the respiring membranes

Having established the clear divergence between the pathways of the solvated proton and of the proton alone, detailed molecular structural data on the respiratory complexes able to handle protons (i.e. complex I, III and IV) are now available thanks to the progress of X-ray analysis and of cryo-microscopy [2–7]. Excellent investigations are available on complex I; here for simplicity we only mention the studies from L. Sazanov and collaborators [2,3].

Numerous structural X-ray studies were conducted on complex I from *Escherichia coli* [4], *Thermus thermophilus* [2] and mammalian ovine (*Ovis aries*) mitochondria [5], and with cryo-microscopy on complex I from *Bos taurus* [6,7]. The complexity of the macromolecular aggregation of complex I is impressive: in mammals it is formed by as many as 45 polypeptides and its assembly needs an unknown number of chaperones, so indispensable that the impairment of just one of them (B17.2 L) causes a progressive encephalopathy [83]. Ogilvie *et al.* [83, p. 2784] state that 'results demonstrate that B17.2 L is a bona fide molecular chaperone that is essential for the assembly of complex I and for the normal function of the nervous system.'

We may seek for the proton plausible pathways inside the respiring membrane, even if it is a plasma-membrane. As only five alpha-helices have been found in the complex I structure, in order to allow for proton translocation, it was necessary to postulate the existence of two hemi channels: one from the matrix (n) side to the centre of the respiratory complex, referred here to as the 'proton entrance hemi channel', and the other from the centre to the periplasmatic (p) side, here indicated as 'proton exit hemi channel'. However, only the latter was well identifiable in complex I [4]. Instead, the entry pathway has not been identified with certainty, so we only can talk about putative pathways labelled with '?' [4]. Furthermore, it clearly emerges from the X-ray studies that there is an obvious proton tunnelling at the centre of the complex I. Comprehensive studies on the protonic movement inside complex I have been carried out by the Helsinki Bioenergetic Group of M. Wikström, which highlighted uncertainty margins on the stoichiometry of protonic extrusion that appear closer to $3\,H^+/2e^-$ [84] instead to the classic $4\,H^+/2e^-$. Also, in the review of Verkhovskaya & Bloch [85] (of Helsinki Bioenergetic Group), four mechanisms for proton translocation are proposed and the 'proton entrance half channel' is not identified with certainty, while the 'proton exit half channel' is clearly identifiable. Emblematically, on the website of the Helsinki Bioenergetic Group it is written (www.biocenter.helsinki.fi/bi/hbg/cl/complex_I_main.html) that 'the mechanism of proton transfer in complex I remains completely enigmatic'.

As far as complex IV (cytochrome *c* oxidase) is concerned, the proton translocation of has been studied in depth [86]. In their recent review, M. Wikström & V. Sharma talk of an 'anniversary': 'Proton pumping by cytochrome *c* oxidase—A 40 year anniversary' [87]. Among the many works cited in this review the *Chemical Review* article of the M. Wikström group stands out [8]. It goes into the details of the possible molecular processes carried out by complex IV [8], thereby including proton translocation. The topic is complex, and more putative pathway of protons are well developed in the review, which we cannot here detail here. For the 'proton entrance half channel', for each molecule of oxygen reduced to water they need $4\,H^+$, and it is hypothesized that another $4\,H^+$ (for a total of 8) enter through this channel. It seems unlikely that there exists equivalence between protons that exist as particles and link to water molecules and protons that should move as a charge, according to the Grotthuss mechanism.

# 9. Proposal for a localized complex I-ATP synthase coupling

Taking into account what is reported above, it is possible to trace a plausible proton pathway within the respiring membrane. In 2006 a direct proton transfer was proposed to couple the respiratory pathway of complexes I, III, IV with ATP synthase [88]. However, in a recent review [21] the concept of transmembrane proton motor force to move the ATP synthase is reinforced, although it is noted that many aspects of coupling are not yet clarified. Clear-cut consideration was proposed some years ago (1991) by Akeson & Deamer [89] about speed of proton translocation through putative proton channel as a limiting step for ATP synthesis by ATP synthase. For the sake of simplicity, we only examine the coupling between the respiratory complex I and the ATP synthase. The existence of a proton pathway at the centre of complex I is quite clear, and we can hypothesize that the protons are sent to the well-identified 'exit half channel', as shown in figure 3. The proton donor at the centre of complex I has already been tentatively identified in the phospholipid cardiolipin (CL) [78], essential for OXPHOS. However, for example, phosphatidylethanolamine (PE) can effectively sustain the functioning of OXPHOS [90]. Data from C. von Ballmoos's group [91,92] unconventionally clarify such a role of phospholipids. In fact, pure phosphatidylcholine (PC) is excellent for the coupling, increasing when the membrane is formed by PC + PE. By contrast, coupling is dramatically inhibited if the membrane is formed by PC + CL, incredibly diverging from the traditional role assigned to CL. This research is also important because experiments are performed with a reconstructed system, more adherent to the 'H$^+$/ATP coupling' or 'second coupling' (figure 1). Here, the driving force that feeds ATP synthase was not the rapid transfer of K$^+$ generated by valinomycin, a widely used method, but the $bo_3$ oxidase of *Escherichia coli*, analogous to the respiratory complex I of vertebrates. It emerges that all membrane phospholipids can act as mobile proton transporters, hindering proton relocation in the near-aqueous medium, so that these are never free. The exergonic process of proton solvation would release an impressive amount of energy (262 400 Cal mol$^{-1}$) [62], which, if not transferred on a generic acceptor, would generate heat devastating not

royalsocietypublishing.org/journal/rsob    Open Biol. 9: 180221

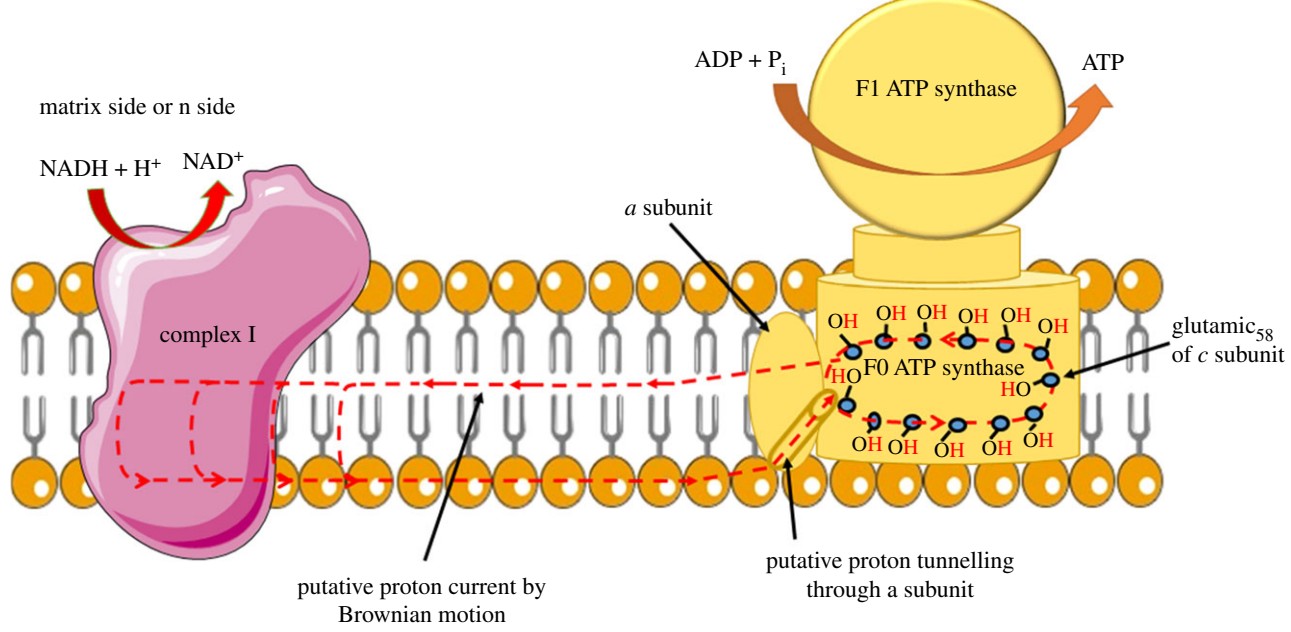

**Figure 3.** A possible H$^+$ circuit inside respiring membrane. The phosphate groups of phospholipids on both sides of the membrane are shown by brown ellipsoids. The image proposes that the H$^+$ (red dotted line) are transferred to the Glu 58 (E58) at the centre of subunit c through subunit a of ATP synthase by proton tunnelling. H$^+$ would flow from the periplasmic side, always bound to phospholipid heads. This can be arranged in each layer of the membrane.

only the membrane but also the cell integrity. Complex I would transfer protons to the p side of the membrane, but these would not be dispersed in the aqueous bulk. In fact it is well documented that there exists a barrier of water molecules attached to the membrane, determining the lateral displacement of protons on the phosphate heads of phospholipids [68,93,94] to meet a sink that is the ATP synthase subunit which, through a proton wire [95,96], would lead the proton to the rotor (figure 3), accommodating on the central glutamic or aspartic residue (depending on the species) of the c subunit, which in variable numbers from 8 to 15 make up the rotor F$_o$ [97]. As F$_o$ rotates it is conceivable that the proton returns to the centre of complex I, a highly hydrophobic environment, thus closing the protonic circuit. This last passage, from the rotor exit to the centre of the respiratory complex, appears plausible for the operativity of the Brownian motion (diffusion) of particles in a highly anisotropic environment that can occur efficiently [45,98] and for a theoretically possible direct passage of the proton exiting the subunit at the entrance to the complexes. Moreover the investigations of the C. von Ballmoos group have produced exhaustive studies with reconstructed systems [91,92,99], in which a direct transfer of proton on the p-side of the membrane from complex I to ATP synthase is evident. Their recent paper also demonstrates that proximity to the membrane between the respiratory complex and ATP synthase is required for the 'proton coupling' [92]. However, the fact remains that the last part of the protonic circuitry (i.e. from F$_o$ moiety of ATP synthase to respiratory complexes) is only hypothetical.

In this controversial scenario, a decisive contribution is undoubtedly the sophisticated bioengineering experiment that labelled both complex IV and ATP synthase with proteins of the GFP family, to experimentally observe a local ΔpH triggered by the respiratory substrate galactose [41]. Observations were conducted in cultured HeLa cells.

The authors concluded [41, p. 1]: 'the observed lateral variation in the proton-motive force necessitates a modification to Peter Mitchell's chemiosmotic proposal'. In other words, the experimentally proven lateral proton motive force is in line with the hypothesis of localized coupling.

## 10. Extramitochondrial oxidative phosphorylation

The chemiosmotic theory [1] as it was formulated envisions a process that can only take place in organelles possessing double membrane systems forming closed compartments to entrap protons, such as mitochondrial cristae, bacteria and thylakoids (here, for the sake of simplicity, we have considered the mitochondrial inner membrane). However, in the last years, several authors have described a functional expression of OXPHOS machinery in cellular districts devoid of mitochondria [100,101], suggesting a possible build-up of a transversal proton gradient across the membrane, out of mitochondria. In particular, an extramitochondrial OXPHOS has been described in rod outer segment (OS) discs [102–105], myelin sheath [79,106–110], cell plasma membrane [111–119], platelets [120], and extracellular vesicles shedding from cells such as exosomes and microvesicles [121,122], which seem to carry an unsuspected metabolic signature [121,123]. Notably, since the plasma membrane potential is positive on the outside and negative on the inside, it would favour ATP hydrolysis rather than its synthesis. The extramitochondrial ATP synthesis is in line with the postulated independence of ATP synthase activity from the membrane potential, and therefore it is plausible that this synthesis of ATP depends on the proton intramembranous coupling.

royalsocietypublishing.org/journal/rsob Open Biol. 9: 180221

# 11. Conclusion

The impressive amount of experimental data cited appears globally in contrast with the above-cited three assumptions underlying the chemiosmotic theory. First of all, it excludes that the protons can accumulate on the coupling membrane surface, whose high permeability would dissipate, since it would correspond to an extreme acidity incompatible with any vital process. Second, since proton diffusion does occur, it is clear that the membrane potential is irrelevant to proton translocation. Third, it can be excluded that the respiratory complexes operate as transmembrane proton transfer from the aqueous bulk, where the proton would exist as hydroxonium ion. This appears to rule out the actual possibility that ATP synthase can overcome the energy barrier, higher than 500 meV [61], required to extract the proton from water, leaving space for the localized coupling hypothesis [63,64], as a dehydration–hydration reaction from hydroxonium ion to free protons would require $262\,400\,Cal\,mol^{-1}$ [62]. Which are the actual processes acting on the ATP synthase? There is no certainty, as highlighted by the emblematic title of an article by J. Walker: 'The ATP synthase: the understood, the uncertain and the unknown' [124].

Indeed, today a constellation of clues leads us to hypothesize the existence of protonic currents internal to the membrane, with the formation of possible circuits travelled by the positive elementary charge, thus realizing a localized coupling that excludes an osmotic nature of the process. To trace this circuit, at least for the possible coupling between respiratory complex I and ATP synthase, it appears realistic that complex I may transfer protons from its central part to the periplasmatic side, allowing them to travel on the membrane surface thanks to the heads of phospholipids [93,94] finally tunnelling inside the a subunit of ATP synthase [95,125] (figure 3). Moreover, several studies show that the membrane is isolated from the aqueous bulk thanks to a layer of water molecules on both sides of the membrane, which consolidates the idea that the membrane is radically distinct and isolated from the liquid phase [59]. Considering the two isolated phases, the only way evolution could pursue to link proton movement to ATP synthesis was a nanomachine connecting the proton movement inside the membrane to the deformation mechanics of the $F_1$ sphere immersed in the aqueous phase. We can reasonably assume that the proton movement inside the membranes occurs as a charge, according to the proton-hopping Grotthuss mechanism, with the establishment of 'protonic currents' inside the membrane [126]. In the non-biological field we find a remarkable adherence to this theory for the development of protonic devices (as proton diodes, transistors, memories and transducers) [77]. It is surprising that the biological and the physical–chemical areas have ignored each other. Since in the history of biology the application of physical–chemical methodologies has led to dramatic advances in biology (we may remind the reader that the resolution of DNA structure [127] was obtained with the fundamental application of X-ray crystallography, developed in 1913 by Williams Henry Bragg and his son Lawrence for the study of inorganic crystals), it is desirable that this fusion of knowledge can be realized in the years to come.

Furthermore, the classical mechanical approach cannot be used to approach these currents. In fact, any time there is a movement of charge bound to a mass, the dualism that cannot be assessed by classical mechanics, instead quantum mechanics must be applied, and in this perspective lay the promising recent quantum-mechanical approaches by Riccardi et al. [76] and Ivontsin et al. [125]. There is a need to take into account the Heisenberg indeterminacy principle as Philip Hunter already highlighted in the title of a 2006 paper: 'A quantum leap in biology: one inscrutable field helps another, as quantum physics unravels consciousness' [128].

Data accessibility. This article has no additional data.

Competing interests. We declare we have no competing interests.

Funding. We received no funding for this study.

Acknowledgements. The authors are very grateful to Professor Giorgio Lenaz, Bologna University (Italy), for the critical reading of the manuscript and for the valuable comments and discussions, and Professor Matthew Stephen Hodgart, University of Surrey, for help in revising the manuscript language.

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
