## [Reviewer comments · Open Biology]

Review History

RSOB-18-0221.R0 (Original submission)

Review form: Reviewer 1

Recommendation

Major revision is needed (please make suggestions in comments)

Are each of the following suitable for general readers?

- a) **Title**
No
- b) **Summary**
No
- c) **Introduction**
No

Is the length of the paper justified?

Yes

Should the paper be seen by a specialist statistical reviewer?

No

Is it clear how to make all supporting data available?

Not Applicable

Is the supplementary material necessary; and if so is it adequate and clear?

No

Do you have any ethical concerns with this paper?

No

Comments to the Author

The submitted manuscript deals with an interesting area of energy transduction and its coupling in biological system. In general, this subject would be interesting for a wide group of readers. Partially, consideration and critical analysis of P. Mitchell theory is certainly very appreciated. However, at the present form the manuscript does not satisfy the aim it has been created for. Instead of comprehensive and detailed analysis the authors made a strong but strange suggestion that the P. Mitchell has to be rejected because non-mitochondrial phosphorylation had been found. This conclusion does not really matter to the theory because the main aspect of chemosmotic theory is a COUPLING MEMBRANE. The general phenomenon is the local energetic barriers near the membrane which create a new possible ability of local coupling. However, this aspect just modifies but does not reject the theory. It needs to be considered and described properly. But it does not.

The text is filled up with many cites of reviews and researches but for a reader it looks like non-systemic. There is no real idea what is a new hypothesis on coupling which is postulated in the title. Extra-mitochondrial phosphorylation is not a new hypothesis of coupling. Possibly, authors need to discuss a parallel existence of proton leakage and transport in the channel. Explain the role of local processes for the coupling. And it will be development but not rejection of Mitchell theory. It will make an appropriate level of knowledge for the readers.

Additionally, the figures in the manuscript are poor and too schematic. If the main aim is to discover the mechanism of coupling at least one figure must contain many details and indications of the process. Thus, the illustrative materials have to be essentially improved.

For minor but essential remarks. In line 236 in the phrase "...transversal or permeability diffusion coefficients..." the word "diffusion" must be omitted. The references are prepared in inaccurate manner. In particular, in line 800 ref.# 117 is formed as a mistake.

Decision letter (RSOB-18-0221.R0)

18-Jan-2019

Dear Dr Morelli,

We are writing to inform you that the Editor has reached a decision on your manuscript RSOB-18-0221 entitled "New hypothesis on coupling sustaining Extra-mitochondrial Aerobic ATP Synthesis", submitted to Open Biology.

As you will see from the reviewer's comments below, there are a number of criticisms that prevent us from accepting your manuscript at this stage. The reviewer suggest, however, that a revised version could be acceptable, if you are able to address their concerns. If you think that you can deal satisfactorily with the reviewer's suggestions, we would be pleased to consider a revised manuscript.

The revision will be re-reviewed, where possible, by the original referees. As such, please submit the revised version of your manuscript within six weeks. If you do not think you will be able to meet this date please let us know immediately.

When submitting your revised manuscript, please respond to the comments made by the referee(s) and upload a file "Response to Referees" in "Section 6 - File Upload". You can use this to document any changes you make to the original manuscript. In order to expedite the processing of the revised manuscript, please be as specific as possible in your response to the referee(s).

Please see our detailed instructions for revision requirements
<https://royalsociety.org/journals/authors/author-guidelines/>

Sincerely,

The Open Biology Team
mailto: openbiology@royalsociety.org

Reviewer's Comments to Author(s):

The submitted manuscript deals with an interesting area of energy transduction and its coupling in biological system. In general, this subject would be interesting for a wide group of readers. Partially, consideration and critical analysis of P. Mitchell theory is certainly very appreciated. However, at the present form the manuscript does not satisfy the aim it has been created for. Instead of comprehensive and detailed analysis the authors made a strong but strange suggestion that the P. Mitchell has to be rejected because non-mitochondrial phosphorylation had been found. This conclusion does not really matter to the theory because the main aspect of chemosmotic theory is a COUPLING MEMBRANE. The general phenomenon is the local energetic barriers near the membrane which create a new possible ability of local coupling. However, this aspect just modifies but does not reject the theory. It needs to be considered and described properly.

The text is filled up with many cites of reviews and researches but for a reader it looks like non-systemic. There is no real idea what is a new hypothesis on coupling which is postulated in the title. Extra-mitochondrial phosphorylation is not a new hypothesis of coupling. Possibly, authors

need to discuss a parallel existence of proton leakage and transport in the channel. Explain the role of local processes for the coupling. And it will be development but not rejection of Mitchell theory. It will make an appropriate level of knowledge for the readers.

Additionally, the figures in the manuscript are too schematic and could be improved. If the main aim is to discover the mechanism of coupling at least one figure must contain many details and indications of the process. Thus, the illustrative materials have to be essentially improved.

For minor but essential remarks. In line 236 in the phrase "...transversal or permeability diffusion coefficients..." the word "diffusion" must be omitted. The references are prepared in inaccurate manner. In particular, in line 800 ref.# 117 is formed as a mistake.

Author's Response to Decision Letter for (RSOB-18-0221.R0)

Our answers are bounded by arrows <--

 New Title "AN UPDATE OF THE CHEMIOSMOTIC THEORY AS SUGGESTED BY POSSIBLE PROTON CURRENTS INSIDE THE COUPLING MEMBRANE" <--

Editor Comments

As you will see from the reviewer's comments below, there are a number of criticisms that prevent us from accepting your manuscript at this stage. The reviewer suggests, however, that a revised version could be acceptable, if you are able to address their concerns. If you think that you can deal satisfactorily with the reviewer's suggestions, we would be pleased to consider a revised manuscript.

 We thank the Editor and the Referee for their considerations and suggestion.

In the revised version, have addressed all the issues raised by the Referee. All the changes made in the text are marked in blue.<--

Reviewer's Comments to Author(s):

The submitted manuscript deals with an interesting area of energy transduction and its coupling in biological system. In general, this subject would be interesting for a wide group of readers. Partially, consideration and critical analysis of P. Mitchell theory is certainly very appreciated. However, at the present form the manuscript does not satisfy the aim it has been created for. Instead of comprehensive and detailed analysis the authors made a strong but strange suggestion that the P. Mitchell has to be rejected because non-mitochondrial phosphorylation had been found. This conclusion does not really matter to the theory because the main aspect of chemiosmotic theory is a COUPLING MEMBRANE. The general phenomenon is the local energetic barriers near the membrane which create a new possible ability of local coupling. However, this aspect just modifies but does not reject the theory. It needs to be considered and described properly.

 We agree with the Reviewer that the main aspect of chemio-osmotic theory is a coupling membrane. In the revised manuscript, we have eliminated the suggestion that the existence of a non-mitochondrial phosphorylation allows to reject the P. Mitchell theory, focusing our attention to the possible local coupling. For the same reason, we have also changed the title of the manuscript in: "AN UPDATE OF THE CHEMIOSMOTIC THEORY AS SUGGESTED BY POSSIBLE PROTON CURRENTS INSIDE THE COUPLING MEMBRANE" <--

The text is filled up with many cites of reviews and researches but for a reader it looks like non-systemic.

 We thank for this suggestion. In the revised manuscript, we have checked the references list and their citation in the text. <--

There is no real idea what is a new hypothesis on coupling which is postulated in the title. Extra-mitochondrial phosphorylation is not a new hypothesis of coupling. Possibly, authors need to discuss a parallel existence of proton leakage and transport in the channel. Explain the role of local processes for the coupling. And it will be development but not rejection of Mitchell theory. It will make an appropriate level of knowledge for the readers.

 We agree with the Reviewer. Indeed, the extra-mitochondrial phosphorylation was cited to justify the idea of proton currents occurring inside the Coupling Membrane instead of across it: considering that the extra-mitochondrial phosphorylation takes place outside the plasma-membrane, free protons could not accumulate there. Therefore, in the revised version, to highlight the crucial existence of proton currents inner the membrane we proposed the following new title : "AN UPDATE OF THE CHEMIOSMOTIC THEORY AS SUGGESTED BY POSSIBLE PROTON CURRENTS INSIDE THE COUPLING MEMBRANE". <--

Additionally, the figures in the manuscript are too schematic and could be improved. If the main aim is to discover the mechanism of coupling at least one figure must contain many details and indications of the process. Thus, the illustrative materials have to be essentially improved.

 We apologize for the poor quality of the images. In the revised version, the three figures were changed, adding more details and indications. <--

For minor but essential remarks. In line 236 in the phrase "...transversal or permeability diffusion coefficients..." the word "diffusion" must be omitted.

 We thank to Reviewer for this suggestion. The word "diffusion" was eliminated in the revised version <--

The references are prepared in inaccurate manner. In particular, in line 800 ref.# 117 is formed as a mistake.

 We apologize for inaccuracy in the references. In the revised version, we have checked and correct the references list. ß

Alessandro Morelli (corresponding Author)
Genova, 14/02/2019 <--

RSOB-18-0221.R1 (Revision)

Review form: Reviewer 1

Recommendation

Accept as is

Are each of the following suitable for general readers?

- a) **Title**
Yes
- b) **Summary**
Yes
- c) **Introduction**
Yes

Is the length of the paper justified?

Yes

Should the paper be seen by a specialist statistical reviewer?

No

Is it clear how to make all supporting data available?

Not Applicable

Is the supplementary material necessary; and if so is it adequate and clear?

Not Applicable

Do you have any ethical concerns with this paper?

No

Comments to the Author

The revised version of the manuscript is improved in general according to the reviewers' comments. Unfortunately, some technical errors still exist in the text (for example, on Page 7 line 29, missed space). Nevertheless, it seems to be in general appropriate for presentation to a common reader. Certainly, many aspects of membrane bioenergetics still remain unclear and they are the points of the discussion. Not all of them included into the review. However, the point of view represented in the submitted manuscript is a general point of possible discussion and it seems to be a background of further development of the coupling theory.

Decision letter (RSOB-18-0221.R1)

13-Mar-2019

Dear Dr Morelli

We are pleased to inform you that your manuscript RSOB-18-0221.R1 entitled "AN UPDATE OF THE CHEMIOSMOTIC THEORY AS SUGGESTED BY POSSIBLE PROTON CURRENTS INSIDE THE COUPLING MEMBRANE" has been accepted by the Editor for publication in Open Biology.

The referee does not recommend any further changes that need to be made. However, we would like to request that the authors carry out a final proof-read of the manuscript and upload the final files for publication.

Please submit the revised version of your manuscript within 14 days. If you do not think you will be able to meet this date please let us know immediately and we can extend this deadline for you.

- 1) A text file of the manuscript (doc, txt, rtf or tex), including the references, tables (including captions) and figure captions. Please remove any tracked changes from the text before submission. PDF files are not an accepted format for the "Main Document".
- 2) A separate electronic file of each figure (tiff, EPS or print-quality PDF preferred). The format should be produced directly from original creation package, or original software format. Please note that PowerPoint files are not accepted.
- 3) Electronic supplementary material: this should be contained in a separate file from the main text and meet our ESM criteria (see <http://royalsocietypublishing.org/instructions-authors#question5>). All supplementary materials accompanying an accepted article will be treated as in their final form. They will be published alongside the paper on the journal website and posted on the online figshare repository. Files on figshare will be made available approximately one week before the accompanying article so that the supplementary material can be attributed a unique DOI.

Online supplementary material will also carry the title and description provided during submission, so please ensure these are accurate and informative. Note that the Royal Society will not edit or typeset supplementary material and it will be hosted as provided. Please ensure that the supplementary material includes the paper details (authors, title, journal name, article DOI). Your article DOI will be 10.1098/rsob.2016[last 4 digits of e.g. 10.1098/rsob.20160049].

- 4) A media summary: a short non-technical summary (up to 100 words) of the key findings/importance of your manuscript. Please try to write in simple English, avoid jargon, explain the importance of the topic, outline the main implications and describe why this topic is newsworthy.

Images

Data-Sharing

It is a condition of publication that data supporting your paper are made available. Data should be made available either in the electronic supplementary material or through an appropriate repository. Details of how to access data should be included in your paper. Please see <http://royalsocietypublishing.org/site/authors/policy.xhtml#question6> for more details.

Data accessibility section

Sincerely,

The Open Biology Team
mailto:openbiology@royalsociety.org

Reviewer's Comments to Author:

Referee:

Comments to the Author(s)

The revised version of the manuscript is improved in general according to the reviewers' comments. Unfortunately, some technical errors still exist in the text (for example, on Page 7 line 29, missed space). Nevertheless, it seems to be in general appropriate for presentation to a common reader. Certainly, many aspects of membrane bioenergetics still remain unclear and they are the points of the discussion. Not all of them included into the review. However, the point of view represented in the submitted manuscript is a general point of possible discussion and it seems to be a background of further development of the coupling theory.

Author's Response to Decision Letter for (RSOB-18-0221.R1)

See Appendix A.

Decision letter (RSOB-18-0221.R2)

22-Mar-2019

Dear Dr Morelli,

We are pleased to inform you that your manuscript entitled "AN UPDATE OF THE CHEMIOSMOTIC THEORY AS SUGGESTED BY POSSIBLE PROTON CURRENTS INSIDE THE COUPLING MEMBRANE" has been accepted by the Editor for publication in Open Biology.

Your application has been approved and no article processing fees apply. You can expect to receive a proof of your article from our Production office in due course, please check your spam

filter if you do not receive it within the next 10 working days. Please let us know if you are likely to be away from e-mail contact during this time.

Sincerely,

The Open Biology Team
mailto: openbiology@royalsociety.org

Appendix A

Response to Referees

Manuscript number: RSOB-18-0221

Original Title "New hypothesis on coupling sustaining Extra-mitochondrial Aerobic ATP Synthesis", submitted to Open Biology.

Our answers are bounded by arrows →←

New Title "AN UPDATE OF THE CHEMIOSMOTIC THEORY AS SUGGESTED BY POSSIBLE PROTON CURRENTS INSIDE THE COUPLING MEMBRANE"

Editor Comments

As you will see from the reviewer's comments below, there are a number of criticisms that prevent us from accepting your manuscript at this stage. The reviewer suggests, however, that a revised version could be acceptable, if you are able to address their concerns. If you think that you can deal satisfactorily with the reviewer's suggestions, we would be pleased to consider a revised manuscript.

→ We thank the Editor and the Referee for their considerations and suggestion.

In the revised version, have addressed all the issues raised by the Referee. All the changes made in the text are marked in blue. And are bounded by arrows ←

Reviewer's Comments to Author(s):

The submitted manuscript deals with an interesting area of energy transduction and its coupling in biological system. In general, this subject would be interesting for a wide group of readers. Partially, consideration and critical analysis of P. Mitchell theory is certainly very appreciated. However, at the present form the manuscript does not satisfy the aim it has been created for. Instead of comprehensive and detailed analysis the authors made a strong but strange suggestion that the P. Mitchell has to be rejected because non-mitochondrial phosphorylation had been found. This conclusion does not really matter to the theory because the main aspect of chemo-osmotic theory is a COUPLING MEMBRANE. The general phenomenon is the local energetic barriers near the membrane which create a new possible ability of local coupling. However, this aspect just modifies but does not reject the theory. It needs to be considered and described properly.

→ We agree with the Reviewer that the main aspect of chemio-osmotic theory is a coupling membrane. In the revised manuscript, we have eliminated the suggestion

that the existence of a non-mitochondrial phosphorylation allows to reject the P. Mitchell theory, focusing our attention to the possible local coupling. For the same reason, we have also changed the title of the manuscript in: “AN UPDATE OF THE CHEMIOSMOTIC THEORY AS SUGGESTED BY POSSIBLE PROTON CURRENTS INSIDE THE COUPLING MEMBRANE” ←

The text is filled up with many cites of reviews and researches but for a reader it looks like non-systemic.

→ We thank for this suggestion. In the revised manuscript, we have checked the references list and their citation in the text. ←

There is no real idea what is a new hypothesis on coupling which is postulated in the title. Extra-mitochondrial phosphorylation is not a new hypothesis of coupling. Possibly, authors need to discuss a parallel existence of proton leakage and transport in the channel. Explain the role of local processes for the coupling. And it will be development but not rejection of Mitchell theory. It will make an appropriate level of knowledge for the readers.

→ We agree with the Reviewer. Indeed, the extra-mitochondrial phosphorylation was cited to justify the idea of proton currents occurring inside the Coupling Membrane instead of across it: considering that the extra-mitochondrial phosphorylation takes place outside the plasma-membrane, free protons could not accumulate there. Therefore, in the revised version, to highlight the crucial existence of proton currents inner the membrane we proposed the following new title : “AN UPDATE OF THE CHEMIOSMOTIC THEORY AS SUGGESTED BY POSSIBLE PROTON CURRENTS INSIDE THE COUPLING MEMBRANE”. ←

Additionally, the figures in the manuscript are too schematic and could be improved. If the main aim is to discover the mechanism of coupling at least one figure must contain many details and indications of the process. Thus, the illustrative materials have to be essentially improved.

→ We apologize for the poor quality of the images. In the revised version, the three figures were changed, adding more details and indications. ←

For minor but essential remarks. In line 236 in the phrase “..transversal or permeability diffusion coefficients...” the word “diffusion” must be omitted.

→ We thank to Reviewer for this suggestion. The word “diffusion” was eliminated in the revised version ←

The references are prepared in inaccurate manner. In particular, in line 800 ref.# 117 is formed as a mistake.

→ We apologize for inaccuracy in the references. In the revised version, we have checked and correct the references list. ←

Alessandro Morelli (corresponding Author)

Genova, 14/02/2019